# Efficient Rigorous Coupled-Wave Analysis Simulation of Mueller Matrix Ellipsometry of Three-Dimensional Multilayer Nanostructures

**DOI:** 10.3390/nano12223951

**Published:** 2022-11-09

**Authors:** Hoang-Lam Pham, Thomas Alcaire, Sebastien Soulan, Delphine Le Cunff, Jean-Hervé Tortai

**Affiliations:** 1LTM, CNRS, CEA/LETI-Minatec, Grenoble INP, Institute of Engineering and Management, Université Grenoble Alpes, 38054 Grenoble, France; 2STMicroelectronics, 38920 Crolles, France

**Keywords:** Mueller matrix ellipsometry, RCWA, 3D multilayer nanostructures, scattering matrix

## Abstract

Mueller matrix ellipsometry (MME) is a powerful metrology tool for nanomanufacturing. The application of MME necessitates electromagnetic computations for inverse problems of metrology determination in both the conventional optimization process and the recent neutral network approach. In this study, we present an efficient, rigorous coupled-wave analysis (RCWA) simulation of multilayer nanostructures to quantify reflected waves, enabling the fast simulation of the corresponding Mueller matrix. Wave propagations in the component layers are characterized by local scattering matrices (s-matrices), which are efficiently computed and integrated into the global s-matrix of the structures to describe the optical responses. The performance of our work is demonstrated through three-dimensional (3D) multilayer nanohole structures in the practical case of industrial Muller matrix measurements of optical diffusers. Another case of plasmonic biosensing is also used to validate our work in simulating full optical responses. The results show significant numerical improvements for the examples, demonstrating the gain in using the RCWA method to address the metrological studies of multilayer nanodevices.

## 1. Introduction

Mueller matrix ellipsometry is a powerful measurement for both academic and industrial applications. By measuring all polarizing states of the sample, this non-destructive measurement is very sensitive to optical responses and has been extensively used for characterizing metrological structures [1,2,3,4,5,6]. The nature of MME is an indirect experimental measurement that necessitates the inverse problem to extract the expected information (grating dimensions, layer thicknesses, optical indices) from the Mueller matrix [7]. The inverse problem is conventionally implemented by the optimization approach, which involves a long iterative process of fitting simulated and measured Mueller matrices [8]. On the contrary, in the recent neural network (NN) approach, well-trained NN models have been reported to achieve high performances in the prediction of metrological geometries [9,10,11]. However, when considering NNs for metrology purposes, a large synthetic dataset must be generated to train the NN model due to the lack of available experimental data. In both optimization and NN approaches, a fast and accurate simulation of nanostructures is critical in order to reduce the computational cost of facilitating the inverse problems of metrology characterization.

Among the numerical methods for periodic nanostructures, RCWA has been widely applied because of its simplicity and accuracy [12,13,14,15]. Since it is well-formulated, RCWA method has been extensively studied for speeding up the modeling of the optical properties of subwavelength-grating structures and improving the efficiency of inverse problem-solving for accurate dimensional determination. The representative studies include the diffractive interface theory to bypass the eigen decomposition of ultrathin metasurfaces [16], the perturbation theory to reduce eigen problems of non-lamellar layers [17,18,19], and normal vectors to improve simulation convergence [20,21]. It is worth noting that these studies were formulated and developed from theoretical structures composed of a representative grating layer. The theoretical structures enable the simulation of multilayer nanodevices because the RCWA method treats component layers independently (Figure 1).

However, this feature may limit the RCWA method when it comes to multilayer nanostructures. In practical applications of biosensing [22], optical diffusers [23], solar cells [24], and photodetectors [25], nanostructures are composed of top gratings and bottom homogeneous layers for tuning the efficiency of the optical devices. Simulating such multilayers involves computing s-matrices of gratings and homogeneous layers, then combining these multiple s-matrices into a global one [26,27] to quantify the expected optical responses. Although the computation of a homogeneous layer is simple, without solving the eigen decomposition, the handling stack of homogeneous layers could be a numerical issue in 3D structures as these layers involve expensive matrix algebra in order to connect to the grating layers. In this study, we present the efficient RCWA simulation of 3D multilayers with a stack of bottom homogeneous layers. We introduce vector-based computation to quickly simulate and integrate s-matrices of homogeneous layers into the global s-matrix. In the RCWA simulation, we also apply a recursive algorithm with bottom-up construction [27] to reduce the components of the global s-matrix. It is noted that the bottom-up construction is presented in decades to enhance the RCWA method by involving only a quarter of the global s-matrix to quantify reflected waves. However, the application of this algorithm is nontrivial. The explanation is provided through the demonstration of modeling the Mueller matrix of the 3D multilayer nanohole structure.

## 2. RCWA Simulation

The RCWA method is well-established, and its numerical implementations have been released [14,28,29,30]. However, the studies usually applied a framework investigated for gratings for a global structure that may decrease numerical efficiency. In this section, we present our RCWA simulation as an efficient solution for multilayers. The distinct feature of our approach is the vector-based formation, which is capable of the fast computing of homogeneous layers in multilayer nanostructures.

### 2.1. Local s-Matrices

#### 2.1.1. Grating Layer

The RCWA method involves the semi-discretization of Maxwell ’s equations in Fourier space to compute the s-matrix of a grating layer. For example, the electric field of a layer is expressed as:(1)$$\frac{\partial^{2}}{\partial z^{2}}e-k_{0}^{2}\Omega e=0$$
where k_0_ is the free-space wave number, and **e** presents the electric field of the layer in Fourier space. Matrix Ω in Equation (1) is defined by:(2)$$\Omega =\left[\begin{array}{cc} K_{x}{⟦\varepsilon ⟧}^{-1}K_{x}⟦\varepsilon ⟧+K_{y}^{2}-⟦\varepsilon ⟧ & K_{x}{⟦\varepsilon ⟧}^{-1}K_{y}⟦\varepsilon ⟧-K_{x}K_{y}\\ K_{y}{⟦\varepsilon ⟧}^{-1}K_{x}⟦\varepsilon ⟧-K_{x}K_{y} & K_{y}{⟦\varepsilon ⟧}^{-1}K_{y}⟦\varepsilon ⟧+K_{x}^{2}-⟦\varepsilon ⟧ \end{array}\right]$$

Here, ⟦ε⟧ is used to describe the permittivity distribution in Fourier space on the x-y plane. **K**_x_, **K**_y_ are diagonal matrices characterizing the wave vector components. (The derivations of Equations (1) and (2) are presented in Appendix A). In this study, matrix Ω in Equation (2) is used to secure the general characteristics of the diffraction. One can quantify the diffraction of s- and p-polarization light by the submatrices $\Omega_{11},$ $\Omega_{22}$ and the cross-conversion of these polarization modes in the submatrices $\Omega_{12},$ $\Omega_{21}$**.** In the studies of the RCWA method [31,32] for 2D structures, the incident plane perpendicular to the y-direction zeroed wave component in the y-direction (**K**_y_ = 0) was applied, consequently resulting in vanished $\Omega_{12}$ $\Omega_{21}$. There is no cross-conversion between the two polarization modes. Thus, the diffraction of s- or p-polarization light can be characterized independently. Either $\Omega_{11}=K_{x}{⟦\varepsilon ⟧}^{-1}K_{x}⟦\varepsilon ⟧-⟦\varepsilon ⟧$ or $\Omega_{22}=K_{x}^{2}-⟦\varepsilon ⟧$ was separately presented in the formulation of the RCWA method for each polarization.

The s-matrix of the grating layer is composed of square submatrices. The submatrices determining reflected and transmitted waves are defined as ([26]):(3a)$$S_{11}={\left(A-XBA^{-1}XB\right)}^{-1}\left(XBA^{-1}XA-B\right)$$
(3b)$$S_{21}={\left(A-XBA^{-1}XB\right)}^{-1}\left(XA-XBA^{-1}B\right)$$
where **X**, **A**, **B** are related to eigenvalues Λ and eigen matrix **W** of matrix Ω. The formulas of these matrices are presented in Appendix B. Equations (3a) and (3b) involve matrix computation with a matrix size of $4N_{h}^{2}$. Herein, N_h_ is the harmonic number in Fourier expansions, and $N_{h}=\left(2m_{x}+1\right)\left(2m_{y}+1\right),$ where m_x_, m_y_ are diffraction orders in the x- and y-directions, respectively.

#### 2.1.2. Homogeneous Layer

For homogeneous layers, a simple computation of the s-matrix without solving the eigen problem is implemented with the advantage of the simple form of permittivity distribution ⟦ε⟧ in Fourier space. ⟦ε⟧ geometrically depends on critical dimensions in the x-y plane. In the simple case of the homogeneous layer, ⟦ε⟧ is a diagonal matrix and related to dielectric property ε by:(4)⟦ε⟧=Iε
with **I** being the unit matrix. Inserting Equation (4) into Equation (2), we obtain a simplified form of matrix Ω:(5)$$\Omega =\left[\begin{array}{cc} K_{x}^{2}+K_{y}^{2}-I\varepsilon & 0\\ 0 & K_{x}^{2}+K_{y}^{2}-I\varepsilon \end{array}\right]$$

In Equation (5), the submatrices $\Omega_{11}=\Omega_{22}=K_{x}^{2}+K_{y}^{2}-I\varepsilon$ are diagonal matrices that entail the form of a diagonal matrix for Ω, eliminating the eigen problem as eigenvalues are elements of the diagonal matrix and eigenvectors have a form of the unit matrix.

#### 2.1.3. Vector-Based Formation

Equations (3a) and (3b) are also used to determine the s-matrix of homogeneous layers and may result in expensive matrix algebra. For 3D subwavelength gratings, the harmonic number N_h_ is very large for accurate RCWA simulation [33], increasing the numerical burden for the computing of the stack of homogeneous layers, though it does not require eigen decomposition. To alleviate the issue, our approach introduces vector-based formation to circumvent the large matrix computation of homogeneous layers. On the right-hand side of Equations (3a) and (3b), **X** is a diagonal matrix and **A**, **B** are square matrices [32]. However, matrices **A** and **B** own a special form composed of four sub-diagonal matrices (Figure 2), which is applied to enhance the RCWA simulation of homogeneous layers. Our approach compactly presents such matrices by the vector of four diagonal elements. Then, the computation of these matrices with the size of $4N_{h}^{2}$ can be implemented fast by only relating to the diagonal elements with a size of $4N_{h}$ (the details are presented in Appendix C).

### 2.2. Global s-Matrix

The global s-matrix is used to quantify the optical responses of the nanostructure. The global s-matrix can be established by connecting multiple s-matrices of component layers. In the numerical implementation, the global s-matrix is computed with the recursive algorithm. The conventional top-down construction, from the top (superstrate) to the substrate, defines the global s-matrix $S^{G}$ [26] as:(6)$$S^{G_{i}}=S^{G_{i-1}}⊗S^{i}$$
where ⊗ is the Redheffer Star Product; the superscript *i* indicates the *i*-th component layer ($i=\bar{1,N}$ with N the number of layers in the nanostructure). **S**^i^ expresses the s-matrix of layer I, and $S^{G_{i}}$ characterizes the global s-matrix computed at iteration *i* or the *i*-th partially collective structure. With this notation, $S^{G_{N}}$ (or **S**^G^ in the reduced form) is the final global s-matrix of the nanostructure. Initially, $S^{G_{0}}$ is the s-matrix of the superstrate that surrounds the sample. All the elements of $S^{G_{i}}$ are obtained in the following expressions:(7a)$$S_{11}^{G_{i}}=S_{11}^{G_{i-1}}+S_{12}^{G_{i-1}}{\left[I-S_{11}^{i}S_{22}^{G_{i-1}}\right]}^{-1}S_{11}^{i}S_{21}^{G_{i-1}}$$
(7b)$$S_{12}^{G_{i}}=S_{12}^{G_{i-1}}{\left[I-S_{11}^{i}S_{22}^{G_{i-1}}\right]}^{-1}S_{12}^{i}$$
(7c)$$S_{21}^{G_{i}}=S_{21}^{i}{\left[I-S_{22}^{G_{i-1}}S_{11}^{i}\right]}^{-1}S_{21}^{G_{i-1}}$$
(7d)$$S_{22}^{G_{i}}=S_{22}^{i}+S_{21}^{i}{\left[I-S_{22}^{G_{i-1}}S_{11}^{i}\right]}^{-1}S_{22}^{G_{i-1}}S_{12}^{i}$$

It is noted that practical applications only require a particular interest in the $S_{11}^{G_{i}}$ and $S_{21}^{G_{i}}$ elements of the global s-matrix, as these elements are respectively used for characterizing reflectance and transmittance (Figure 3). Equation (7a) shows that computing $S_{11}^{G_{i}}$ requires the four exact values of the full matrix $S^{G_{i-1}}$. In other words, the construction in Equation (6) necessitates the full global s-matrix for modeling reflectance.

In order to reduce the submatrices of the global s-matrix, the computation can start from the bottom to the top. With the rearrangement of the Redheffer Star Product, the bottom-up construction of the global s-matrix is expressed as ([27]):(8)$$S^{G_{i}}=S^{N-i}⊗S^{G_{i-1}}$$

In this construction, the initial $S^{G_{0}}$ is the s-matrix of the substrate. The submatrices of interest $S_{11}^{G_{i}}$ and $S_{21}^{G_{i}}$ are defined as:(9a)$$S_{11}^{G_{i}}=S_{11}^{N-i}+S_{12}^{N-i}{\left[i-S_{11}^{G_{i-1}}S_{22}^{N-i}\right]}^{-1}S_{11}^{G_{i-1}}S_{21}^{N-i}\;$$
(9b)$$S_{21}^{G_{i}}=S_{21}^{G_{i}}{\left[i-S_{22}^{N-i}S_{11}^{G_{i}}\right]}^{-1}S_{21}^{N-i}$$

Only two equations are needed instead of the four required for the top-down construction. Equation (9a) shows that $S_{11}^{G_{i}}$ is uniquely related by itself and layer s-matrices in numerical computation. The advantage of bottom-up construction is illustrated in Figure 3. Only half of the global s-matrix is required to simulate full optical responses, and only a quarter of the global s-matrix is involved in determining the reflectance, enabling the fast simulation of the corresponding Mueller matrix.

Generally, $S_{11}^{G_{i}}$ and $S_{21}^{G_{i}}$ in Equations (9a) and (9b) involve large matrix algebra with the matrix size of $4N_{h}^{2}$. However, by starting from the substrate, $S_{11}^{G_{i}}$ and $S_{21}^{G_{i}}$ also possess a special form composed of four sub-diagonal matrices. Thus, vector-based computation also facilitates integrating homogeneous layers into the structures in numerical work.

## 3. Numerical Demonstration

### 3.1. Case 1: Mueller Matrix Ellipsometry

We validate and discuss the performance of our work with the industrial case: MME of the 3D multilayer nanohole structure. The structure is composed of five layers: the first two nanohole patterns of Si_3_N_4_, SiO_2_ are stacked on three following homogeneous layers: Si_3_N_4_, SiO_2_, and Si_3_N_4_, respectively. In our case, these nanoholes are metrology dies on STMicroelectronics wafers [34] in order to control the process for imaging devices (diffusive patterns). Experimental MME was measured at an angle of incidence of 65° and azimuth of 45° and provided by STMicroelectronics, Crolles, France. The cross dimension of the hole was measured at 120 nm using an inline CD-SEM (ATLAS system from Onto Innovation).

We use our laboratory measurement for the refractive indices of Si, SiO_2_ (shown in Figure 4) and the Tauc-Lorentz single oscillator dispersion model [35] for the dielectric function of Si_3_N_4_, the parameters of which are given in Table 1:

RCWA is implemented by our in-house code for the numerical demonstration of the nanohole structure (Figure 5a). The source code is publicly accessible [36]. Matrix Ω in Equation (5) is used for computing the local s-matrices of the three bottom homogeneous layers. For Si_3_N_4_/SiO_2_ nanohole layers, solving the eigen problem of matrix Ω in Equation (2) is required for these layer s-matrices. Three RCWA algorithms are implemented for the comparison. Conventional RCWA needs a full global s-matrix to model the reflected waves (Equations (7a)–(7d)). RCWA with bottom-up construction only involves a quarter of the global s-matrix (Equation (9a)). Our work applies both bottom-up construction and vector-based formation for the fast solution of modeling the multilayers. The Mueller matrix is calculated from the global s-matrix with zeroth-order reflected light (Appendix D).

### 3.2. Case 2: Full Optical Responses

Another multilayer nanohole pattern is considered to test our approach. The structure consists of an Al nanohole pattern on three homogeneous layers of SiO_2_, Si, and Ge, respectively (Figure 5c). Such a structure is widely applied in biosensing as its plasmonic resonance is sensitive to the refractive index change induced by targeted biological molecules [22]. In numerical implementation, the optical responses are simulated in deionized water at normal incidence, with the incident plane perpendicular to the y-direction. The documented refractive indices of Al [37], Ge [38], and deionized water [39] were selected in the computation. Similar to the demonstration in Case 1, we employ Equations (2) and (5) for the s-matrices of the homogenous layers and nanohole pattern, respectively. To obtain full optical responses, $S_{11}^{G},\;S_{21}^{G}$ from Equations (9a) and (9b) are used to calculate the reflectance (R) and the transmittance (T). The absorbance (A) is determined by: A = 1 − R − T. The vector-based formation is applied to the computation related to homogeneous layers.

It is noted that similar multilayers were also studied in [30] to demonstrate the advantage of the RCWA method over the finite-difference time domain (FDTD) in designing plasmonic applications. Their numerical results are used as a reference for the validation of our work.

## 4. Results and Discussions

### 4.1. Mueller Matrix Ellipsometry

The simulation time of the three algorithms is normalized by the total simulation time of our work and is provided in Table 2. The enhancement is presented in percentages defined by the ratio of the algorithms. For simplicity, we only calculate the ratio needed for the discussion. Compared to bottom-up and conventional RCWA, this work accelerates the computation by 1.51 and 1.91 times, respectively. The bottom-up algorithm optimizes the RCWA method by only computing reflected waves, resulting in a significant reduction of 70% in computing the global s-matrix. However, it is worth remembering that solving layer s-matrices is still the key issue in the RCWA method, highly affecting numerical computations. Consequently, the bottom-up algorithm only saves 21% of the total computation time compared to the conventional RCWA. It can be the reason that the bottom-up algorithm is nontrivial for the application of multilayers in metrological studies.

Although computing homogeneous layers can be fast without solving the eigen problem, it is also time-consuming to handle large matrices in order to connect these layers to grating layers. In conventional and bottom-up RCWA, the normalized time of three homogeneous layers lasts 0.4, which is about 43% of the time needed for two Si_3_N_4_, SiO_2_ nanohole patterns. This ratio dramatically declines to less than 1% in our work. The impressive improvement results from our approach that circumvents large matrices by using representative vectors for fast computation. It also reduces the 88% computational time of the global s-matrix. With these improvements, our work saves 48% of the total simulation time.

The Mueller matrix is calculated from reflected waves. As shown in Figure 6, the simulated Mueller matrix is in good agreement with the industrial measured one of the demonstrated structure. This result confirms the accuracy of the RCWA method, which is favorable for the metrological studies of 3D nanomultilayers.

### 4.2. Full Optical Responses

Figure 7a presents the full optical responses with observed Fano resonance, e.g., at 1290 nm of the absorbance spectrum of the demonstrated nanomultilayers. Figure 7b shows the simulation time of the two RCWA algorithms. In all studied diffraction orders, our work achieves higher numerical efficiency. Accurate performance of the RCWA method necessitates high diffraction orders, which exponentially increase simulation time. For tasks of understanding important optical responses, a diffraction order m_x_ = m_y_ = 5 can be favorable for acceptable accuracy. A higher value, e.g., m_x_ = m_y_ = 10, should be considered for more rigorous demand in applications of inverse design in metrological investigations. In these instances, our work is about 10 times and 2 times faster, respectively.

However, it must be noted that the comparison is relative as the numerical efficiency may result from differences in computational frameworks, such as the programming languages, simulation targets, and computer resources. For example, ref. [30] employed Julia in a node of a workstation cluster with 24 cores, while our work uses Python within a personal laptop equipped with Intel^®^ core i7 CPU to perform numerical computations. In this section, we demonstrate that the numerical investigation of the multilayers can be conducted in a reasonable time using bottom-up construction and vector-based computation. Our algorithm mainly deals with the matrix algebra of only one patterned layer instead of all four layers, which entails high performance in the demonstration.

Lastly, it should be pointed out that the impressive improvement in the two demonstrated cases of Muller matrix modeling and full optical responses is obtained without intervening with the eigen problem of the grating layers. Thus, our approach is easily implemented in the RCWA method. Owing to the compelling enhancement, simplicity, and accuracy, our work is useful to the community as similar structures with a large number of homogeneous underlayers are widely used in optical applications.

## 5. Conclusions

Practical applications of multilayer nanostructures with manipulated wave propagations require efficient numerical simulations for targeted optical responses. By introducing the vector-based formation of fast computing and integrating homogeneous layers into nanostructures, we present the simple but efficient RCWA simulation of multilayers with bottom homogeneous layers. An impressive performance is obtained in the demonstration of the 3D nanomultilayers in both the industrial MME measurement of optical diffusers and the full optical response of plasmonic structures. Thus, our work is expected to serve as a numerical simulation approach in future work for the metrological studies of engineered diffraction applications.

## Figures and Tables

**Figure 1 nanomaterials-12-03951-f001:**
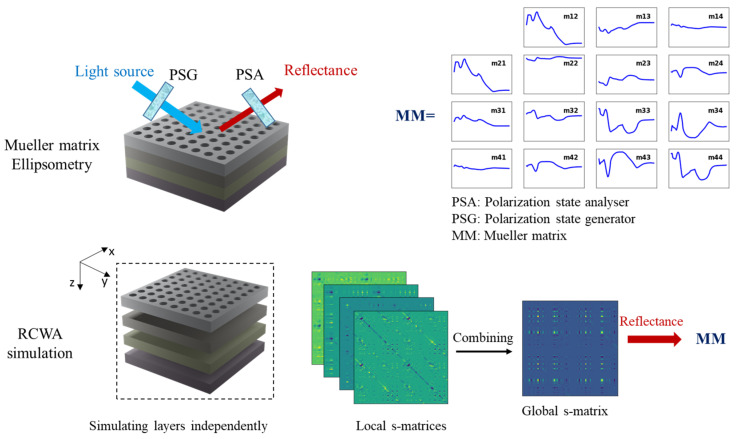
Schematic of Mueller matrix ellipsometry and RCWA simulation of multilayers. By measuring all polarizing states of reflected waves, the metrological information of the multilayers is contained in the Mueller matrix (MM). In the RCWA method, local layers are independently simulated and combined for the global s-matrix to quantify reflectance to compute MM in a wide range of measured wavelengths.

**Figure 2 nanomaterials-12-03951-f002:**
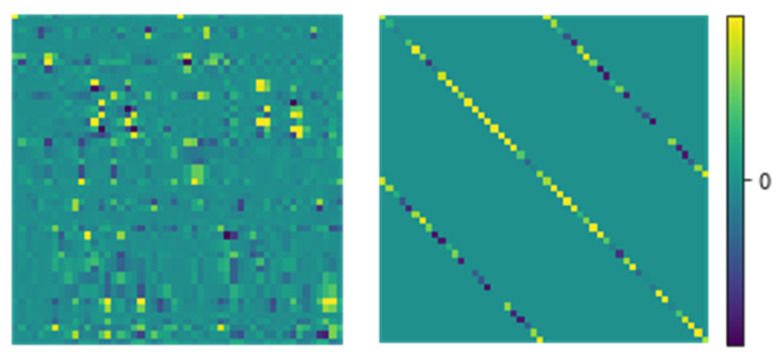
Example of the matrix form in the computation of local s-matrices in the RCWA method. On the left is a completed matrix form involved in grating layers. On the right is the special form composed of four sub-diagonal matrices, which is applied to enhance the RCWA simulation of homogeneous layers.

**Figure 3 nanomaterials-12-03951-f003:**
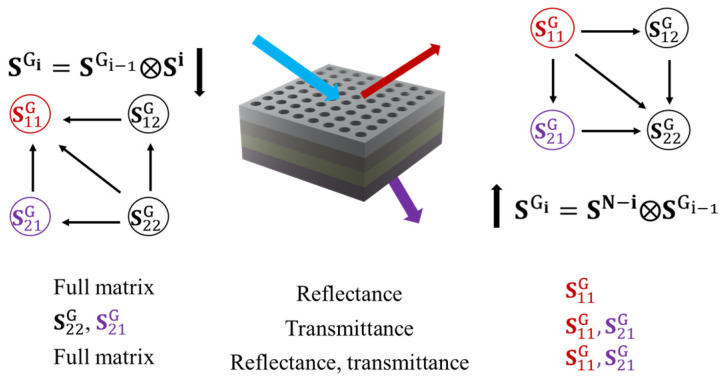
Recursive algorithm of constructing the global s-matrix, with the computational network representing the involvement of elements. The color nodes are the main interest, as they are used for characterizing optical responses. The bottom space represents the advantages of bottom-up versus top-down construction.

**Figure 4 nanomaterials-12-03951-f004:**
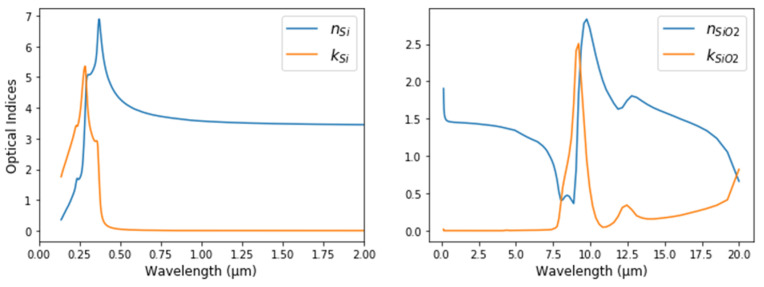
Optical indices of Si and SiO_2_ used in the numerical demonstration.

**Figure 5 nanomaterials-12-03951-f005:**
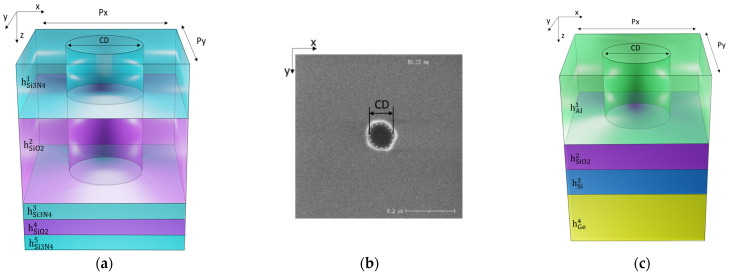
(**a**) Schematic of the unit cell of the Si_3_N_4_/SiO_2_ nanohole structure in Case 1. (**b**) Scanning electron microscopy (SEM) cross-section image in the x-y plane of the unit cell of the Si_3_N_4_/SiO_2_ nanohole. (**c**) Schematic of the unit cell of the Al nanohole pattern in Case 2. The dimension in the two schematics is not to scale.

**Figure 6 nanomaterials-12-03951-f006:**
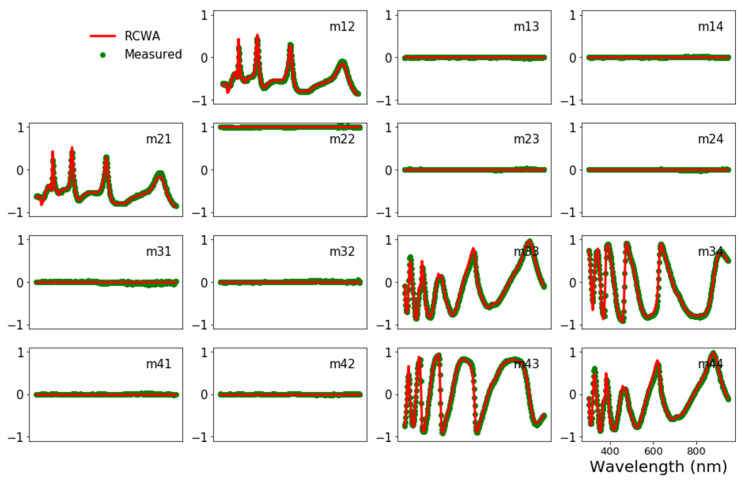
Mueller matrix of measured data (green solid circle) and RCWA simulation (red solid line). The parameters of the structure in the simulation are: P_x_ = P_y_ = 470 nm, CD = 120 nm, $h_{\mathrm{Si}3N4}^{1}=50\;\mathrm{nm}$, $h_{\mathrm{SiO}2}^{2}=660\;\mathrm{nm}$, $h_{\mathrm{Si}3N4}^{3}=50\;\mathrm{nm}$, $h_{\mathrm{SiO}2}^{4}=80\;\mathrm{nm}$, $h_{\mathrm{Si}3N4}^{5}=22\;\mathrm{nm}$, angle of incidence = 65°, azimuth angle = 45°.

**Figure 7 nanomaterials-12-03951-f007:**
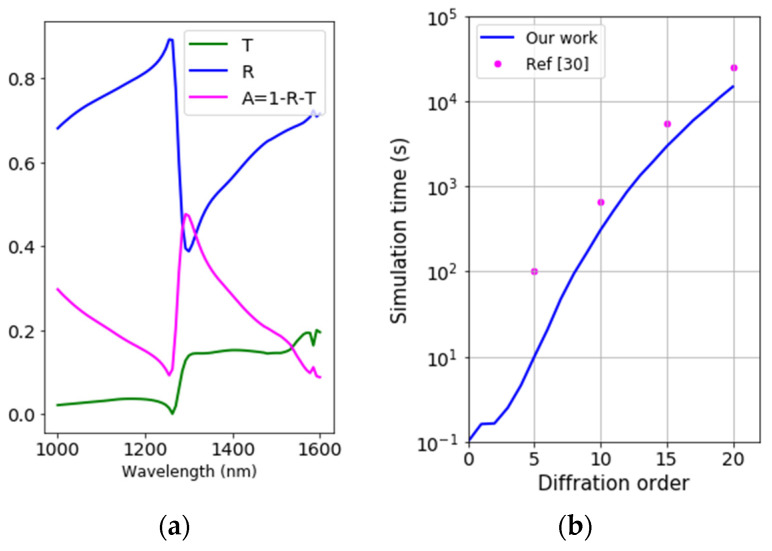
(**a**) Simulated spectra of the plasmonic nanostructure, using 81 wavelength points from 1000 to 1600 nm, P_x_ = P_y_ = 950 nm, CD = 500 nm, $h_{\mathrm{Al}}^{1}=100\;\mathrm{nm}$, $h_{\mathrm{SiO}2}^{2}=50\;\mathrm{nm}$, $h_{\mathrm{Si}}^{3}=50\;\mathrm{nm},$ and $h_{\mathrm{Ge}}^{4}=1000\;\mathrm{nm}$. (**b**) RCWA simulation time of both studies as a function of diffraction order in x- and y-directions.

**Table 1 nanomaterials-12-03951-t001:** Tauc-Lorentz parameters used to describe the dielectric function of Si_3_N_4_.

Parameters	Symbol	Value (eV)
Oscillator amplitude	$A_{\mathrm{TL}}$	104.27
Peak energy	$E_{\mathrm{TL}}^{0}$	8.22
Broadening term	$C_{\mathrm{TL}}$	3.38
Optical band gap	$E_{\mathrm{TL}}^{g}$	4.25
Real part of the dielectric function at infinite energy	$\varepsilon_{\mathrm{TL}}^{\infty }$	1.64

**Table 2 nanomaterials-12-03951-t002:** Normalized simulation time of the 3D multilayer nanohole structure.

Simulation	Conventional	Bottom-Up	This Work	Bottom-Up/Conventional	This Work/Conventional
Total time	1.91	1.51	1	79%	52%
Grating layers	0.93	0.93	0.93	-	-
Homogeneous layers	0.4	0.4	0.0026	-	-
Homogeneous layers/Grating layers	43%	43%	0.27%	-	-
Global s-matrix	0.5	0.15	0.06	30%	12%

## Data Availability

The source code of our simulation is publicly accessible [36].

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
