# Peer review of "Efficient Rigorous Coupled-Wave Analysis Simulation of Mueller Matrix Ellipsometry of Three-Dimensional Multilayer Nanostructures"

_nanomaterials, 2022, doi:10.3390/nano12223951_

Round 1
Reviewer 1 Report (Previous Reviewer 1)
I appreciate the efforts made by the authors to correct the original draft.
The current revision provides a clearer vision of the study presented by the authors.
Despite still being affected by a certain lack of novelty due to the analytical aspects placed at the center of this work, now the draft presents characters which make it appropriate to publishing.
Author Response
Thank you very much for the constructive comments.
Reviewer 2 Report (New Reviewer)
GENERAL REMARKS
This manuscript is written on a topic that has a large significance in both theory and applications. The text and the figures are of high quality. The code is publicly available and validated by other methods. These make the impact of the work even more significant. I suggest acceptance after considering a few comments as detailed below.
MAJOR COMMENTS
1. Equations (1) and (2): the orientation of the x, y and z, coordinates marking the propagation is missing. Maybe a coordinate system could be inserted, even if only into Fig. 1.
2. It would be nice to demonstrate (even plot) the differences between the numerical results of a conventional RCWA calculation or, e.g., FDTD calculation and the improved RCWA calculation. Are there implications on the sensitivity of parameters or convergence?
3. It would be informative to see a list of the fitted parameters with values and uncertainties (cross-correlations).
4. Were the simulated spectra of Fig. 7a verified by measurements? It would be a nice demonstration to compare it with transmission/reflection measurements.
5. Was multi-threading applied? It has probably a much larger influence on the speed, because running the code on a 24-core computer, as mentioned in the manuscript, could be at least an order of magnitude faster by implementing multi-threading in the python code. I guess python cannot deal with this out of the box, although the problem is easily parallelizable due to the independence of the calculation of the individual spectral points.
NOTES
Line 222: similar multilayers was > similar multilayers were
Author Response
Please see the attachment

This manuscript is a resubmission of an earlier submission. The following is a list of the peer review reports and author responses from that submission.
Round 1
Reviewer 1 Report
The manuscript "Efficient rigorous coupled-wave analysis simulation of Mueller matrix ellipsometry of three-dimensional multilayer nanostructures" of H.L. Pham et al. presents a detailed procedure to get a fast evaluation of the S-matrix in multilayered optical systems combined with a final scattering layer, and consequently to calculate the corresponding Mueller matrix tipically used in ellipsometry analysis; the S-matrix of the complete system is obtained throughout the RCWA protocol starting from the S-matrices of each compounding individual layer. Within the RCWA process, the final S-matrix evaluation is speeded up by primarily considering just 2 coefficients out of the total 4 for their more relevant roles in finding the scattering coefficients, and by assuming an inverted matrix combination protocol. This helped the authors to get a faster evaluation of the Mueller matrix for each sample configuration of the multilayered system, and so to enhance the efficiency of a numerical setup developed within a neural network framework for a massive analysis on a huge collection of configurations.
The authors provides the reader with details of the algorithm and show the results calculated for a reference structure at optical frequencies, together with a comparison of the calculation efficiency both by means of the conventional analysis, of the ordinary recursive method based on the more canonical S-matrix derivation, and by their own method, all expressed in terms of processing times.
Despite being useful for a theoretical researcher doing accurate numerical evaluations of the scattering process on optical structures, this work is affected by lack of originality because of the core topic here discussed, and by the absence of a clear description of the internal stages of the efficiency enhancement evaluation.
Indeed, system matrix evaluation protocols for multilayered optical structures have been at the center of theoretical investigations for decades, and many papers and books have already been published each presenting useful methods for the matrix evaluation enhancement; this is also vaguely recognized by the authors along lines 70-72 of the manuscript. Furthermore, the increase in efficiency, being here expressed as the process time ratio in a comparison with more ordinary methods, is declared in the text by stating some time values without the internal description of the numerical stages compounding the overall analysis process; this second point is crucial, because process time is a very complicated matter strictly related to the multi-stage nature of optical systems simulations.
Therefore, I consider this work at its current stage to need more elements before being worthy of publication; I kindly encourage the authors to find more aspects of the S-matrix calculation protocol for the improvement, to perform the comparison of the evaluation processes with high care of details and present all of them in the draft, and then reconsider again their work for publication. Aside this, there are some statements in the current manuscript to adjust for their misleading character or improper English grammar. Here it follows a list of these minor corrections:
- Line 78: “We present RCWA method with…..” -> maybe it should be ” We describe the RCWA method as….”
- Line 120-121: Please, use a better phrase in replacement of “we introduce the approach to optimize the advantage of homogeneous layers”.
- Line 144: maybe the “global structure” here cited should better be referred to as the i-th partially collective structure.
- Line 152-153: Please, recast “only focus on global s-matrix with a particular interest in”; a good candidate could be “(practical applications) only require particular interest in S11 and S21 of the global S-matrix”.
- Line 160: please, fix the sentence “With the construction different from”.
- Line 203: “Which is based on” could be improved in “which is based on the procedure described in references”.
- Table 2: we do not know the time unit used for the data, and the row indicated by “Homogeneous layers/ Grating layers” only reports percentages rather than time values.
- Line 234: please, fix “It should be pointing that”; maybe it could be “It should be pointed out that”.
- Line 251: Please, remove the “…” and complete the statement.
Reviewer 2 Report
Report on:
Efficient rigorous coupled-wave analysis simulation of Mueller matrix ellipsometry of three-dimensional multilayer nanostructures
by
Hoang-Lam Pham, Thomas Alcaire, Delphine Le Cunff, Sebastien Soulan, Jean-Hervé Tortai
The paper is devoted to the study of the Mueller matrix ellipsometry by the use of the RCWA (rigorous coupled-wave analysis). In particular, the paper introduce an efficient computational approach for performing RCWA simulations for multi-layer nanostructures.
To this end, local S-matrices are calculated for each of the plane layers of the multi-layered structure and used to obtain the global S-matrix of the whole structure. This is allowing for a fast computation of the global S-matrix, and from it, the Mueller matrix of the structure. The suggested approach is claimed to open up for more realistic use of inverse scattering problem techniques to characterize system based on measured Mueller matrix responses. The performance of the proposed approach is demonstrated by applying it to a multi-layer nanohole structure.
This topic is currently of significant scientific interest and any progress to the field is very welcome due to the numerous applications that potentially may benefit from its advances.
Even if the paper represents an interesting research topic, my main criticism of the current version of the manuscript is the following:
1. Overall, I find it hard to follow the paper. The presentation I do not find very clear and the used symbols are not always defined and/or explained after they are first introduced. This makes it difficult, or maybe even impossible for readers, to reproduce the results presented in the paper. It is suggested that the authors present their approach in a way that is more accessible to the readers and define ALL the quantities used just after introducing them. Currently, this is not the case, and the manuscript suffers from not doing so! For instance, in the discussion around Eqs. (1) and (2):
- What is the meaning of $\Gamma$
- How are the matrices $P$ and $Q$ for a given geometry defined; this is not clear to me.
- What is meant by $[[\varepsilon ]]$; It is said in the text to be the permitivity distribution --- what is this meaning? Is it a matrix and if so, what is the dimension.
- what are the dimensions of the matrices involved.....
- What is the meaning of $\mathbf{K}_x$ and $\mathbf{K}_y$ etc
2. Is it a rigorous result that the global S-matrix can be constructed by tensor products according to Eq. (9). If any approximation are needed to arrive at this results, it needs to be explained in detail. I suspect that certain multiple scattering effects are not included when such a procedure is used, but I may be wrong. Anyhow, it is interesting to explain this in detail and to maybe refer to relevant literature on the topic.
- in the example the authors apply their approach to, the three bottom layers are planar and the global S-matrix is constructed according to Eq. (9) by a bottom-up approach. Could alternatively one use a top-down approach? What is the nano-holes were in the two layers just after the substrate?
3. Many results are just listed without any reference and/or explanation. Here the authors should give references to where the interested reader can go in order to get further information. For instance, the basis of the RCWA method is said to relay on Eq. (1). Here one or several references are in order. Furthermore, after solving Eq. (1) how is the S-matrix obtained?
4. Finally, the manuscript has language issues throughout the paper which makes it difficult to understand and read. The authors are advised to carefully work on the language and/or to get professional help to improve it.
Minor issues:
* There should not be a new paragraph (no indent) after most of the equations. E.g. after Eqs. (1)--(4)
* Eq. (1) The meaning of \Gamma is not explained in the text.
* Why do the authors use the term "s-matrix" (lower case) when a more common phrase is "S-matrix"; at least, this is what I am most familiar with seeing.
* There are typos in Eqs. (6) and (7) regarding the use of boldface.
For all the above reasons, I can not recommend the publication of the manuscripts in its current form. In my view, the current version of the paper has simply too many issues. The paper has potential and if the criticism is dealt with properly I welcome a revised version of the paper.
Reviewer 3 Report
Please find the attached document for comments

Round 2
Reviewer 2 Report
Report on the revised version of the manuscript:
Efficient rigorous coupled-wave analysis simulation of Mueller matrix ellipsometry of three-dimensional multilayer nanostructures
by
Hoang-Lam Pham, Thomas Alcaire, Delphine Le Cunff, Sebastien Soulan, Jean-Hervé Tortai
I have now gone over the answers to my original report and the revised version of the manuscript. Still, I find that the paper is not very clear and it remains hard to read. The rather few changes to the manuscript have not changed that.
Hence, I recommend that the manuscript is rejected.
Reviewer 3 Report
I have no further comment. This manuscript can be accepted.